# A Classification Method for Workers’ Physical Risk

**DOI:** 10.3390/s23031575

**Published:** 2023-02-01

**Authors:** Christian Tamantini, Cristiana Rondoni, Francesca Cordella, Eugenio Guglielmelli, Loredana Zollo

**Affiliations:** Research Unit of Advanced Robotics and Human-Centred Technologies, Università Campus Bio-Medico di Roma, 00128 Rome, Italy

**Keywords:** worker risk prevention, physiological monitoring, fall prediction

## Abstract

In Industry 4.0 scenarios, wearable sensing allows the development of monitoring solutions for workers’ risk prevention. Current approaches aim to identify the presence of a risky event, such as falls, when it has already occurred. However, there is a need to develop methods capable of identifying the presence of a risk condition in order to prevent the occurrence of the damage itself. The measurement of vital and non-vital physiological parameters enables the worker’s complex state estimation to identify risk conditions preventing falls, slips and fainting, as a result of physical overexertion and heat stress exposure. This paper aims at investigating classification approaches to identify risk conditions with respect to normal physical activity by exploiting physiological measurements in different conditions of physical exertion and heat stress. Moreover, the role played in the risk identification by specific sensors and features was investigated. The obtained results evidenced that k-Nearest Neighbors is the best performing algorithm in all the experimental conditions exploiting only information coming from cardiorespiratory monitoring (mean accuracy 88.7±7.3% for the model trained with *max(HR)*, *std(RR)* and *std(HR)*).

## 1. Introduction

About 84% of all non-fatal injuries and illnesses leading to days away from work in 2020 involved slips, trips, falls, overexertion or exposure to harmful substances or environmental conditions, according to International Labour Organisation (ILO) statistics [1,2]. With the advent of the Industry 4.0 paradigm, worker monitoring to understand their state during the working day is crucial in order to prevent and identify risk factors before they can lead to actual damage to health [3]. Among all the aforementioned injury causes, the phenomenon of falling is the one that is most analyzed in the literature.

Accelerometers, gyroscopes, pressure sensors, video/depth cameras, microphones and radio frequency sensors are sensor modalities that are generally used to perform fall detection [4,5]. The accelerometers are considered the gold standard since they are unobtrusive and the signal analysis allows identifying an impact by using simple threshold-based approaches [6] as well as more sophisticated machine learning algorithms [7,8]. Although scientific research has extensively studied the problem of fall detection, the methods used are capable of detecting a fall event caused by the impact of the body on the ground when it has already occurred [9,10]. They are not suitable for preventing or reducing the severity of injuries caused by the impact itself [11]. Developing algorithms capable of identifying risky conditions before they lead to a harmful event, such as a fall, is necessary.

Physiological monitoring may represent an effective way to collect information about workers’ health and safety statuses [12]. Cardiorespiratory activities, Galvanic Skin Response (GSR) and Skin Temperature (ST) have emerged as the physiological parameters that allow estimating users’ exertion [13,14]. Supervised approaches based on k-Nearest Neighbors, Decision Tree and Linear Discriminant Analysis have been proposed, both binary [15] and multiclass [16], to estimate the subjective perception of users’ physical workload while performing their working activities starting from physiological measurements. Moreover, physiological parameters reflect the heat stress applied to the user [17,18]. In particular, Heart Rate (HR) is significantly impacted by environmental conditions [19]. Indeed, physiological parameters exhibit responses not only according to what the users are doing, i.e., their level of physical workload [20,21], and the environment they are in, i.e., exposure to heat stress, but also reveal information about the responses of the autonomic nervous system, related with risk awareness. When people face significant risk, the sympathetic system of the autonomic nervous system generates responses measurable from vital and non-vital parameters. Indeed, these responses can be related to the presence of a certain risk factor. Physiological parameters can feed a supervised learning method based on a Support Vector Machine (SVM) to estimate the stress levels of construction site workers generated by risk factors in the workplace [22]. On the other hand, this study needed invasive information from the users, since the data were labeled on the basis of the cortisol level in the blood of the workers enrolled in this study. Wearable non-invasive instrumentation can be used to highlight risk conditions. Starting from GSR measurement, low- and high-risk activities, manually labeled by a human operator, can be distinguished [23]. It has not been demonstrated whether it is possible to discriminate a risk condition from normal physical activity under different conditions of physical fatigue and environmental conditions of heat stress. It is worth investigating whether physiological information that can be retrieved by means of wearable unobtrusive instrumentation could serve as valuable inputs to identify risks.

This work aims at proposing the best method of achieving risk identification starting from physiological sensing. For this purpose, several classifiers already used in the literature are compared in order to measure their performance in identifying a risky condition between two activities that are difficult to distinguish using conventional accelerometer-based approaches. Moreover, since the physiological parameters are influenced by the activity of the user and the environment, an experiment is designed to collect data in different physical workloads and heat stress conditions. The presence of physical risk of falling was then simulated by means of a stabilometric platform while a treadmill was used to induce physical exertion in the participants.

The rest of the paper is structured as follows: Section 2 describes materials and methods that are used in this study. The experimental setup used to monitor the physiological parameters along with the experimental protocol to physically place participants under the different environmental conditions is also explained. The obtained results are presented and discussed in Section 3. Lastly, Section 4 draws the conclusions of the study and provides future work.

## 2. Materials and Methods

This section presents the proposed approach to identify risky conditions shown in Figure 1. Specifically, a monitoring system is used to collect data during an experiment in which participants are exposed to different levels of physical exertion and environmental conditions characterized by different heat stress. A Physiological Monitoring System (PMS) along with a Movement Monitoring System (MMS) monitor physiological and acceleration data. The collected information is monitored and pre-processed to extract and select the most relevant features for the risk identification problem faced in this paper. The computed features are given as input to supervised learning classifiers to train and test machine learning algorithm performance in predicting risk. All the blocks are described in depth in the following.

### 2.1. Physiological Monitoring System

The PMS proposed in this work includes measurements of cardiorespiratory activity, GSR and ST, as these were found to be the parameters that most closely reflect the activities of the autonomic nervous system in response to external stimuli, such as risk factors [24].

The electrocardiogram provides multiple pieces of information about the health and cognitive state of the user. Heart Rate (HR) and Heart Rate Variability (HRV) are the two main features that can be extracted from the electrical activity of the heart. Indeed, HR exhibits modifications according to the activity performed and the health status of the user. For instance, HR decelerates after the stimuli administration, and a high varying HR with respect to the baseline value is associated with high aroused conditions [25]. On the other hand, the HR increases and the HRV decreases whenever the user performs an intense physical activity [26].

Many disorders and/or stimulus administrations generate alterations in respiratory activity. Respiration Rate (RR) is a vital sign sensitive to different pathological conditions and/or stressors including emotional stress, cognitive load, heat, cold, physical effort and exercise-induced fatigue [27]. Moreover, it can be used to assess the psychophysiological state of a user even if its modifications are much slower than those exhibited by other physiological parameters.

GSR is a physiological parameter commonly used to assess users’ cognitive state [28]. The higher the sweating, the higher the increase in the electrical conductance of the skin. Hands and feet are commonly used to measure GSR since they exhibit the highest density of sweat glands in the body. Moreover, the GSR is made of two different components—the Skin Conductance Level (SCL), the tonic and slowly changing part—reflecting the participants’ arousal and the Skin Conductance Response (SCR), the phasic fast-changing one, reacting to stimulus administration.

ST is another important parameter for determining the psychophysiological state and can be used to predict heat stroke conditions [29]. In fact, sudden or excessive increases in the users’ temperature may reveal abnormalities. It may depend on some individual factors (i.e., age, gender) and external factors [30].

### 2.2. Movement Monitoring System

Data about the acceleration of a user can be retrieved by means of M-IMUs sensors [31]. Such sensors quantify the movements of the body districts where the sensors are located. In this work, the accelerations of the trunk and head are monitored to monitor the user’s motion. These anatomical landmarks are chosen as they are the most used in fall detection algorithms [32].

### 2.3. Feature Extraction and Selection

Several parameters can be extracted from the recorded raw signals. The ECG allows the computation of HR and HRV. The HR is defined as the number of R-peaks per minute in the ECG trace. Given the Inter-Beat Intervals ($IBI_{H}$), defined as the time intervals between consecutive heartbeats, it is possible to compute the instantaneous *HR*, expressed in beats per minute (bpm), as [33]:(1)$$HR\left(i\right)=\frac{60}{IBI_{H}\left(i\right)}.$$

The Root Mean Square of Successive Differences (*RMSSD*) is a time domain HRV metric, computed as:(2)$$RMSSD\left(i\right)=\sqrt{\frac{1}{N}\sum_{i=1}^{N}{\left(IBI_{H}\left(i\right)-IBI_{H}\left(i+1\right)\right)}^{2}}$$
where *N* represents the number of $IBI_{H}$ in the sequence.

From the breathing waveform, respiratory events are detected by identifying the local maxima which represent the maximum expansion of the rib cage. As for the $IBI_{H}$ computation, an inter-breath interval signal ($IBI_{R}$) can be defined. The *RR*, expressed as breath per min (bpm), can be computed as
(3)$$RR\left(i\right)=\frac{60}{IBI_{R}\left(i\right)}.$$

The GSR requires signal processing to retrieve the tonic and phasic components. They can be extracted by applying a 5 Hz Butterworth low pass filter, to remove noise and motion artifacts, and then applying a 0.1 Hz low pass Butterworth filter to retrieve the SCL component. On the other hand, a 0.1 Hz high pass Butterworth filter returns the SCR component. It is worth assessing GSR peaks, defined as increases of a minimum of 0.03–0.05 µ(µS) in the SCR, since they enclose information about the administered stimulus [34].

All the collected data require a normalization procedure to allow comparisons. In fact, physiological features exhibit high intra- and inter-subject variability as a result of age, gender, time of day and many other factors. Normalization reduces the effect of such variability by evaluating the response of each physiological parameter with respect to the baseline value, i.e., collected in a rest condition of the user. In particular, the participants are asked to sit comfortably, blindfolded and acoustically isolated for 5 min. The normalization is performed as:(4)$$x_{\mathrm{norm}}=\frac{x-x_{\mathrm{baseline}}}{x_{\mathrm{baseline}}}$$
where *x* is the physiological signals acquired at a given time stamp, $x_{\mathrm{baseline}}$ is the mean value computed in the baseline recording and $x_{\mathrm{norm}}$ is the normalized physiological vector.

The M-IMUs placed on the anatomical landmarks of the users return accelerations around the three axes. As a synthetic measure, the Acceleration signal Vector Magnitude (*AVM*) is computed as:(5)$$AVM=\sqrt{a_{x}^{2}+a_{y}^{2}+a_{z}^{2}}$$
where $a_{x}$, $a_{y}$ and $a_{z}$ are the measured accelerations along the $\vec{X}$, $\vec{Y}$ and $\vec{Z}$ axes of the M-IMU, presented in Figure 2.

The feature extraction should be computed over a temporal window that allows capturing appreciable variation in the signals of interest. In particular, the physiological signals slowly change. Therefore, the time window should be of the order of seconds [35,36]. A time window of 2.5 s is used to extract statistical features from the collected data. Such a time window has been set since it is also used for the fall detection algorithm based on classification of acceleration data [37]. The mean, standard deviation (*std()*), minimum (*min()*), maximum (*max()*), mean of the first derivative (*fd()*) and mean of the second derivative (*sd()*) are computed in the time window. Given the SCR is a zero-mean signal, the absolute value of the signal is considered during feature extraction. The computation of the statistical feature from the {HR,HRV,RR,SCL,SCR} leads to 36 physiological features. Moreover, additional features are extracted from SCR such as the number of peaks (*N_pks*), average peak amplitude (*mean(A_pks)*), the standard deviation of peak amplitude (*std(A_pks)*) and the energy of the GSR signal response (*E_gsr*) [38]. In particular, the GSR energy is computed as:(6)$$E_{GSR}=\int_{-\infty }^{+\infty }{\left|F_{GSR}\right|}^{2}df$$
where
(7)$$|F_{GSR}{|}^{2}=F_{GSR}\cdot F_{GSR}{}^{*}$$
is the auto spectrum of the signal given $F_{GSR}$, the Fast Fourier transform of the GSR signal. Since there are two M-IMU sensors returning three acceleration values along with its AVM, statistical features were calculated for the movement monitoring system. Ultimately, 40 physiological features and 24 movement features were extracted from the collected data to build up the PHY and ACC datasets, respectively.

From all the features that have been extracted, the weights of the features that make a positive contribution to the classification have been quantified in this paper. In order to highlight the most informative features that can be used to highlight risk perception, an automatic feature selection method is applied to the PHY dataset. The ReliefF algorithm, along with its derivatives, is an evaluation filter algorithm capable of detecting feature dependencies by using the concept of nearest neighbors to derive feature statistics that indirectly account for interactions [39]. The ReliefF method is based on associating a score with each feature in the dataset, which can be used to rank the features from best to worst performing. Taking a labeled sample in the dataset and a feature, a weight is calculated according to the *k*-nearest sample belonging to the same class (near hits) and the *k*-nearest sample belonging to another class (near misses). If the distance to the near hits is greater than the distance to the near misses, the weight associated with a feature increases; otherwise, it decreases. The number of samples *k* is empirically set. Taking into account reduced feature sets may increase the classifiers’ performances.

### 2.4. Supervised Model for Risk Assessment

Four machine learning algorithms are selected to cope with the current classification task. They are k-Nearest Neighbors (kNN), Linear Discriminant Analysis (LDA), Gaussian Support Vector Machine (SVM) and Decision Tree (Tree). kNN is one of the most used machine learning algorithms used to process physiological data [40]. LDA is one of the simplest machine learning methods effective in many classification problems in which physiological signals are involved [41]. SVM turned out to be an algorithm suitable for fatigue identification [42]. Decision Tree algorithms have already been used for the identification of fatigue conditions from physiological signals [16]. A brief description of the algorithms is reported in the following:k-Nearest Neighbors (kNN): This computes the prediction according to the majority of the K-nearest patterns in data space. The hyper-parameters characterizing the algorithm behavior are the number of neighbors, i.e., the number of samples of the known class to be considered closest to establish the class of the unknown sample and the metrics used to compute the distance among the samples.Linear Discriminant Analysis (LDA): This model aims at identifying a hyperplane separating the elements of different classes. It fits a Gaussian density to each class, assuming that all classes share the same covariance matrix.Gaussian Support Vector Machine (SVM): This method finds a non-linear separation between samples in a transformed high dimensional feature space. The algorithm performance is mainly influenced by the mathematical functions that can be used to transform the observations before assigning the prediction. Such space transformation is used to highlight a Gaussian-like separation between the samples belonging to different classes.Decision Tree (Tree): The objective of Tree is to create a model that predicts the target value through simple decision rules derived from the features.

The aforementioned models were implemented in MATLAB R2020b software using the hyperparameters presented in Table 1.

The accuracy, defined as the proportion of the total number of correct predictions with respect to the total number of tested samples, is used to assess the classification performance. Accuracy is a metric that generally describes how the model performs across all classes. It is useful when all classes are of equal importance and the dataset has a balanced number of observations among classes [43]. Moreover, the time needed to train the models and infer a prediction is measured.

### 2.5. Experimental Setup

The experimental setup used in this study is composed of the commercial sensing devices shown in Figure 2.

The MMS is composed of two MTw Xsens Inertial Measurement Units (IMU) that are used to acquire movement information. One is placed on the chest by means of a corsage worn by the participant and another is fixed on the head with an elastic band.

The PMS consists of three sensory systems to monitor vital and non-vital physiological parameters. The Zephyr BioHarness^TM^ chest belt is used to measure cardio-respiratory activity. It is a wireless wearable device to be worn in direct contact with the skin, which allows real-time recording of physiological parameters related to the cardio-respiratory system, including HR, HRV and RR [44]. The Shimmer3 GSR + Unit is used to measure the GSR between two reusable electrodes attached to the proximal phalanx of the index and middle fingers, respectively. The ST is recorded by means of the Shimmer Skin Surface Temperature Probe, whose sensible area is fixed on the chest of the participants and connected to the Shimmer3 Bridge Amplifier+ unit. The chest is chosen as the anatomical landmark for sensor placement since it turned out to be one of the best positions to estimate the internal body temperature from skin measurement [45].

The acquisition of all the sensors was temporally synchronized and collected at 40 Hz by using the *Yet Another Robotic Platform* (YARP) [46].

### 2.6. Experimental Protocol

An experiment, involving healthy participants, was carried out in order to validate the possibility to identify risk under different physical workloads and heat stress conditions. The Biodex stabilometric platform, presented in Figure 3A, was used to generate a 1 min of risk condition, i.e., the risk of falling by balance loss. In this trial, the base of the platform is unlocked to disrupt the balance of the participants. The non-risky activity that was carried out by the volunteers is doing physical activity on the Walker View treadmill, see Figure 3B.

In order to generate two different levels of physical exertion in the participant, the treadmill tasks did not have a fixed duration but conditions were defined that had to be fulfilled in order for the task to be completed. In particular, the participants were asked to walk and/or run on the treadmill until their HR reached a percentage of their critical HR ($HR_{c}$) [20], defined as
(8)$$HR_{c}=\left\{\begin{array}{c} 220-age\;\;\;\;\;\mathrm{for}\;\mathrm{males}\\ 206-0.88\cdot age\;\mathrm{for}\;\mathrm{females} \end{array}\right.$$
where, as evident, a different $HR_{c}$ can be computed according to the participant’s age and sex. Two critical thresholds are defined and set to complete the task: 60% and 85% of $HR_{c}$. Once the participant current HR exceeds the threshold, the treadmill task can be concluded. The speed of the first walking task is set at 5 km/h at 6% inclination. In the second repetition, the speed is set at 6 km/h with a 9% of inclination. Speed and inclination are chosen because the angle of the treadmill produces a high physical workload [47].

Moreover, the full protocol is repeated in two different environmental conditions, characterized by different heat stress exposure levels. To objectify the effect of the environment on participants, the Environmental Risk Coefficient (*ERC*) [20] is computed as
(9)ERC=T+RH·0.1
where *T* is the environmental temperature in Celsius degrees and RH is the relative humidity. For ERC values ≤29, the workers can carry out their activities without any concern. In environmental conditions characterized by 30≥ERC>34, the heat begins to cause dehydration so the worker should drink frequently. For higher values of the environmental risk coefficient (ERC≥34), workers are exposed to an increasing heat risk factor and should avoid high-intensity work. The ORIA wireless thermometer–hygrometer is used to measure both the temperature and relative humidity of the environment during the experiments. The instrument records environmental data every 10 s and sends them to a mobile phone application. Such a device measures temperature ranging from −20 °C to 60 °C with a ±0.5 °C accuracy and humidity range from 0 to 99% *RH* with a ±1% RH. In more detail, the two acquisition sessions were carried out in the following environmental conditions:E1: temperature of 22.19±0.55 °C and 44.58±1.97% humidity;E2: temperature of 28.81±0.53 °C and 52.80±4.94% humidity.

In the tested environmental conditions, the risk coefficients are $ERC_{E1}=26.64\pm 0.57$ and $ERC_{E2}=34.08\pm 0.72$ in *E1* and *E2*, respectively. This means that the two environmental conditions are properly exposing the participants to different heat stress levels.

Eight healthy participants, 4 males and 4 females with mean age of 24.0±2.6 years, were enrolled in this study. The recruited volunteers were informed about the procedure and equipped with the monitoring system. At the beginning of the experiment, the participants were asked to rest, blindfolded and acoustically isolated for 5 min in order to collect the physiological baseline of each subject, paramount in the physiological data normalization in Equation (Equation 4).

For each of the tested environmental conditions, the participants underwent an initial test on the stabilometric platform. The participants were then asked to walk until they reached the first $HR_{c}$ threshold, i.e., 60%, and immediately afterward they repeated the task on the platform. The volunteers then underwent the second phase of physical exertion up to the 90% of their $HR_{c}$ and, lastly, they repeated the risk condition on the stabilometric platform. Data collected during the experiment were labeled as “RISK” and “NON-RISK” as the participant performed the proposed tasks.

### 2.7. Algorithm Validation

The Leave-One-Subject-Out (LOSO) validation method was used to validate the proposed classification algorithms. Having *N* individuals, this method consists of cyclically using N−1 participants to train the machine learning algorithms and test the performance of the trained model on the remaining one.

At first, the four machine learning classifiers were trained and tested by using all the data collected from the accelerometers and the physiological sensors, i.e., the ACC and PHY datasets, respectively. This first validation step serves to assess the actual need to use physiological information to identify risk perception. This first validation step serves to assess the actual need to use physiological information to identify risk perception. Fall detection approaches are in fact already able to identify when a fall has occurred, but the aim of this work is to assess the performance of classifiers in recognizing when users are subjected to the presence of risk before it leads to injury. The rest of the analysis conducted in this study will focus on the PHY dataset.

Secondly, an analysis is carried out to assess the effect of the environmental conditions on the algorithms’ performance. Model accuracy is compared by separating the performance obtained in environment E1 from those in the thermally stressful E2. Indeed, physiological parameters can be altered by environmental conditions, especially heat stress. The classifiers’ accuracy in identifying risk conditions whatever the working environment is assessed.

The specific contribution to the performance of the implemented classifiers is then quantified by calculating the accuracy of the models by modifying sensory information by removing some of it from the PHY dataset. Starting from the full PHY dataset made of GSR, cardiorespiratory (CR) and ST data, seven configurations can be defined: (I) PHY = GSR + CR + ST, (II) GSR + CR, (III) GSR + ST, (IV) CR + ST, (V) CR, (VI) GSR and (VII) ST. Through this analysis, therefore, it will be possible to identify which sensory information has a greater effect on the performance of the classifiers in recognizing risk perception than in the condition in which all physiological information is present.

Lastly, the ReliefF feature selection algorithm will be applied to identify the weight of each feature in risk identification with $k\in \left[1,3,5,10\right]$. Given the weight of each feature, the classifiers will be re-trained on a set of features defined as optimal (Optimal PHY) in which, for each classifier, the *N* highest weighted features returning a higher average accuracy over the eight enrolled participants will be considered.

### 2.8. Statistical Analysis

The validation of the risk identification approach goes through a comparative analysis presented in Section 2.7. For each proposed comparison, Wilcoxon’s non-parametric statistical test was applied to the resulting accuracy. The significance level is set at 0.05. In analyses where multiple comparisons are required, e.g., where one piece of sensory information is removed at a time, the Bonferroni correction is applied [48]. Specifically, the threshold value for *p*-values becomes $0.05/n_{c}$, where $n_{c}$ represents the number of multiple comparisons performed.

## 3. Results and Discussions

At the end of the experimental sessions, 320,000 raw samples of physiological parameters and accelerations were collected. Specifically, 20,000 samples were acquired from each subject under each thermal stress condition. After the feature extraction step, performed on 2.5 s time windows (100 samples), the two datasets ACC and PHY consist of 3200 samples per each feature. The first analysis conducted concerns the comparison of the classification accuracy of the four selected machine learning models using the two datasets collected in the experimental acquisitions: ACC and PHY. Accuracy is shown in Figure 4 as boxplots. In particular, the horizontal line inside the box represents the median value of the accuracy calculated in the LOSO validation on the eight enrolled subjects; the box encloses the interquartile range, while the whiskers show the minimum and maximum values of the data. In addition, isolated points in the graph show outliers, defined as points that fall below the lower quartile − 1.5 times the interquartile range or above the upper quartile + 1.5 times the interquartile range. The outliers were taken into account during statistical comparisons among the tested approaches.

As evident, all machine learning models are more accurate in recognizing risk situations when using PHY information with respect to ACC. In particular, kNN, LDA and Tree performance are significantly higher with *p*-values <0.001, <0.01 and 0.03, respectively. Moreover, it is worth noting that the classifiers trained on the ACC dataset perform very close to 50%, which is representative of a system that identifies binary conditions randomly. This shows how the two conditions tested are indistinguishable from the point of view of the movement, whereas physiological information is able to identify risky conditions. kNN and SVM classifiers emerge as the most accurate and the least performing with median accuracy of 75.9% and 71.1%, respectively. Table 2 presents the time needed to train the proposed approaches ($T_{train}$) and infer a prediction ($T_{pred}$). All classifiers trained with the PHY dataset required training times of the order of a second. Apart from these very short times, the time required to infer a prediction of perceived risk is even more important. Times on the order of tens of milliseconds make it possible to quickly identify risk and generate corrective actions such as alerting the worker to the risk to prevent conditions that could potentially cause harm. For completeness, further metrics of the implemented classifiers are reported in Appendix A, Table A1.

The performance of the classifiers was then detailed by separating the observations collected in the two different environmental conditions, reported in Figure 5.

Heat stress, administered in E2, induces an alteration of the physiological state of the enrolled participants. This effect is reflected in the performance of the implemented machine learning models, which showed a decrease in accuracy. However, this decrease is not significant. In fact, the *p*-values obtained from the statistical tests are 0.8, 0.5, 0.7 and 0.8 for kNN, LDA, SVM and Tree, respectively. This means that machine learning approaches are able to identify perceived risk conditions irrespective of the environmental stresses on the user. Among all the implemented approaches, kNN turned out to be the most robust one, exhibiting the slightest decrease in performance. Precision, recall and F1 scores are presented in Appendix A, Table A2.

Figure 6 shows the sensors’ contribution to the classification performance. In particular, the accuracies obtained by training the models with different sensorial inputs are reported.

The performance of each model trained with the complete PHY dataset was compared with the accuracy obtained by modifying sensory information by removing some of its one-by-one sensory information. kNN turned out to be one of the most sensible models. The performance is significantly degraded in the condition GSR + CR, GSR + ST and GSR with *p*-values of 0.003 for all the comparisons. That means that the GSR features alone are not capable of identifying risk when using the kNN model. The multimodal combination of physiological parameters returned an improved estimation.

LDA is the model that manages to achieve comparable performance in all tested configurations except for the condition where only surface temperature information is used (*p*-value = 0.003). The ST proves to be insufficient in terms of discriminating a possible risk for LDA.

SVM turns out to be the model that exhibited a non-significant performance degradation when modifying sensory information by removing some of its sensorial information. In fact, performance degrades with the deletion of sensory information without significant drops. The performance obtained in the configurations GSR + CR, GSR + ST and GSR obtained median accuracy of 52.9%, very close to the random prediction case.

Lastly, the Decision Tree performance significantly suffers a deterioration in the GSR condition (*p*-value = 0.007).

Among the three sensors used in this experiment, BioHarness seems to be the one that achieves the most accurate predictions even when used alone. This means that the physiological responses obtained by analyzing cardiorespiratory activity are good predictors of risk conditions. In order to gain more detail from this analysis, the ReliefF automatic feature selection method was applied to quantify the specific contribution of each individual feature of the PHY dataset. The results of ReliefF showed that the features with a positive weight are always the same across all the enrolled participants, for all the *k* parameter values selected, i.e., $k\in \left[1,3,5,10\right]$. Figure 7 displays the computed weights per each feature obtained with k=5. The features coming from the BioHarness, the GSR and ST Shimmer sensors are colored in blue, red and green, respectively.

The features that obtained the highest median weights from ReliefF are *max(HR)*, *std(RR)* and *std(HR)* with 0.049, 0.043 and 0.042, respectively. It is even more evident that the features coming from BioHarness, i.e., those related to CR, are the most appropriate with respect to identifying the risk condition studied in this paper. Moreover, analyzing the feature weights extracted from the PHY dataset reveals that there are some features with negative weights. Most of these features are from the GSR analysis. In particular, the statistical features extracted from the SCR exhibit negative weights. Since the SCR is zero mean adjusted in post-processing, this signal does not carry a high information content when analyzing the statistical features. In contrast, *mean(A_pks)* and *std(A_pks)* obtained positive median weights.

Given the weights of each feature, the four machine learning algorithms were trained by including one feature at a time, from the heaviest to the least important. For each model, the number of features with the best median accuracy is selected, defining the optimal set of physiological features. In particular, kNN, LDA, SVM and Tree obtain the best results for 3, 25, 5 and 18 features, respectively. Figure 8 shows the consequent accuracy improvement obtained by training and testing the models only with the Optimal PHY with respect to the initial PHY.

All the implemented models exhibit improved performance when trained only with the optimal set of physiological features. LDA, SVM and Tree improve the medians of performance, although not significantly. In contrast, kNN remains the best model and with only three features achieves an accuracy of 88.7±7.3, significantly improving on the performance 73.4±12.0 showed using the entire PHY dataset (*p*-value = 0.04). Additional information can be found in Appendix A, Table A3.

Among all the models implemented, kNN appears to be the most appropriate for identifying risk conditions. Indeed, it appears to be able to capture knowledge from the observations of the subjects used in the training phase. The physiological responses exhibit highly non-linear behavior that is captured by kNN regardless of the thermal environmental condition to which the participants are subjected. By removing sensory information that does not contribute positively to classification, kNN is able to achieve the best performance among all the implemented models. In particular, cardiorespiratory information alone turns out to be the most suitable for identifying risky conditions. In other words, cardiorespiratory monitoring systems turned out to be the most relevant for integration in intelligent wearable systems for workers. In fact, in the operating environment, it does not appear to be practicable to equip the worker with all the sensory systems used during the laboratory tests carried out in this work.

The performance of the approaches presented in this paper strongly depends on the experimental conditions under which the datasets were acquired and the choice of features extracted from the raw data collected. Using different experimental conditions and/or choosing different feature sets could in fact alter the classification performance. Moreover, the optimal feature set that emerged in this work could change if different features were taken into account in the dataset. Each new predictor could positively as well as negatively affect the classification problem faced in this paper.

## 4. Conclusions

This study presented an analysis to demonstrate the ability of machine learning models to identify risk conditions from physiological information. In contrast to fall-detection algorithms in the literature, where accelerometers are used to identify a fall once it has occurred, in this work awareness of the risk of falling was generated using a stabilometric platform. This condition was studied with respect to different conditions of physical fatigue and thermal stress. In particular, a study was designed and conducted by enrolling eight healthy participants.

The dataset composed of physiological information performed better in recognizing the risk condition than a dataset composed only of M-IMU sensor monitoring head and trunk accelerations. This made it possible to demonstrate how the autonomic responses of the autonomic nervous system generate physiological responses in relation to risk perception. The classifiers proved robust to the presence of environmental heat stress, showing non-significant performance degradation. Furthermore, an analysis was conducted to investigate the effect of removing physiological information by excluding one sensor at a time and applying a feature selection method. The cardiorespiratory information turns out to be the most informative with respect to predicting risk conditions. The kNN algorithm achieved an accuracy of 88.7±7.3% with a minimum feature set consisting of the three highest-ranked features.

Future efforts will be devoted to collecting data from more participants and enrolling a more heterogeneous population. At the same time, exploring different feature sets could be useful for defining further metrics to identify risks. Furthermore, the same algorithmic approaches should be validated by collecting data with sensors integrated within intelligent wearable systems, to be validated in an operational environment.

## Figures and Tables

**Figure 1 sensors-23-01575-f001:**
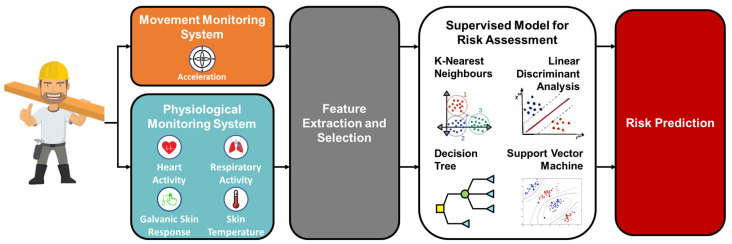
Block scheme of the proposed approach.

**Figure 2 sensors-23-01575-f002:**
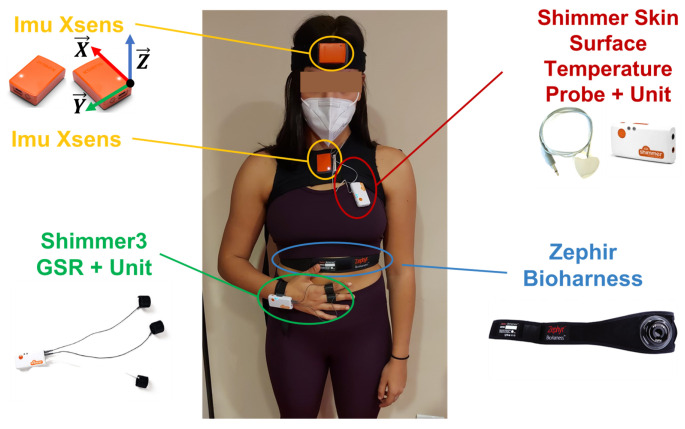
Experimental setup used during the experiments. The local reference frame of the M-IMU sensors is also reported.

**Figure 3 sensors-23-01575-f003:**
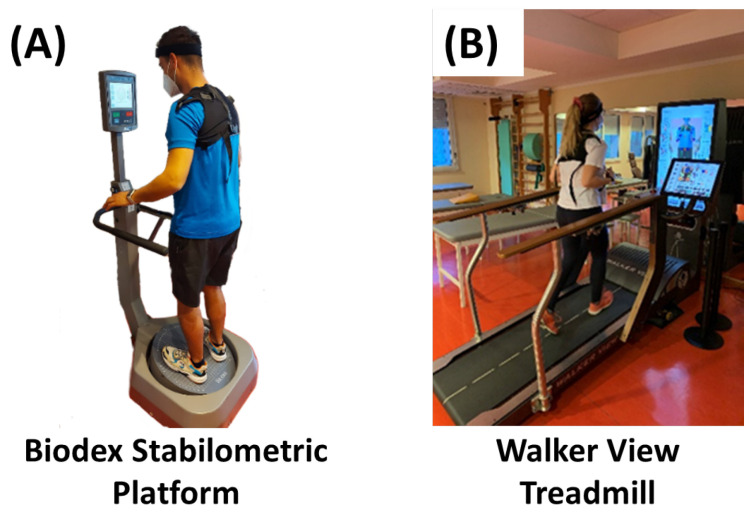
Platform used in the experiments. (**A**) The Biodex stabilometric platform is used to simulate a fall event. (**B**) The Walker View treadmill is used to physically exert the participants.

**Figure 4 sensors-23-01575-f004:**
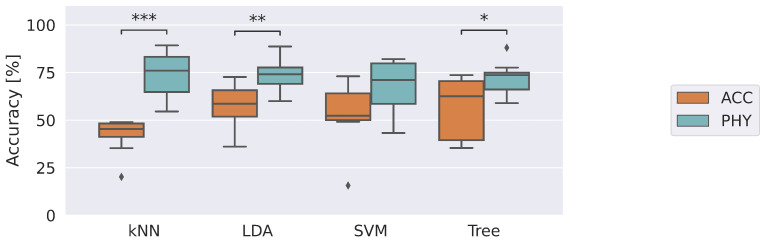
Accuracy returned by the classifiers in identifying risk by using data of accelerations and physiological data, ACC and PHY datasets, respectively. The *, **, and *** denote comparisons in which 0.01<*p*-value ≤0.05, 0.001<*p*-value ≤0.01, and *p*-value ≤0.001, respectively.

**Figure 5 sensors-23-01575-f005:**
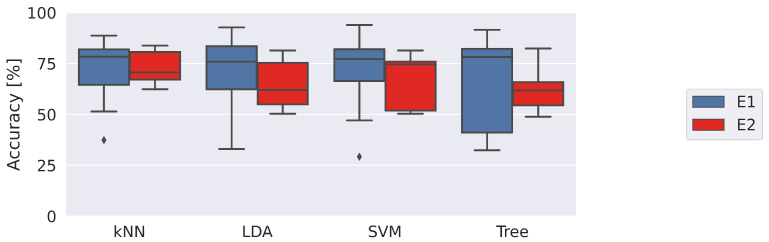
Accuracy returned by the classifiers trained with the PHY dataset in the two tested environmental conditions.

**Figure 6 sensors-23-01575-f006:**
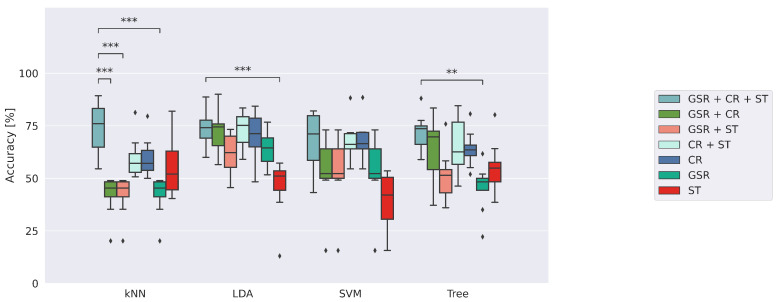
Accuracy of the implemented classifiers trained with different sensorial inputs. The ** and *** denote comparisons in which 0.001<*p*-value ≤0.01 and *p*-value ≤0.001, respectively.

**Figure 7 sensors-23-01575-f007:**
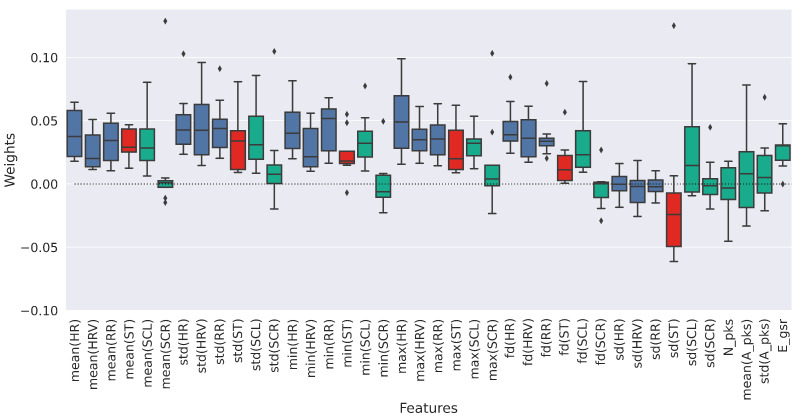
Feature weights computed with the ReliefF algorithm from the PHY dataset of the eight enrolled participants.

**Figure 8 sensors-23-01575-f008:**
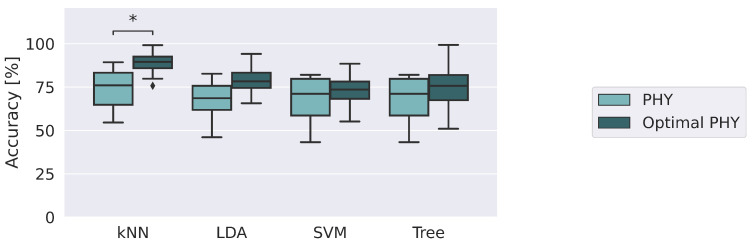
Accuracy returned by the classifiers in identifying risk by using physiological data and the optimal feature set, i.e., PHY and Optimal PHY datasets, respectively. The * denotes comparisons in which *p*-value ≤0.05.

**Table 1 sensors-23-01575-t001:** Hyperparameters of the implemented models.

	HyperParameters	Tested Values
kNN	Number of Neighbors	1
	Distance Metrics	Euclidean
LDA	Discriminant Type	Linear
SVM	Kernel Function	Gaussian
	Kernel Scale	1
	Box Constraint Level	1
Tree	Split Criterion	Gini’s Diversity Index
	Predictor Selection Algorithm	Standard CART
	Minimum Branch Node Observations	10
	Minimum Leaf Node Observations	1

**Table 2 sensors-23-01575-t002:** Time to train the proposed approaches and infer a prediction.

	$T_{\mathrm{train}}$ (s)	$T_{\mathrm{pred}}$ (s)
kNN	0.08±0.17	0.024±0.021
LDA	0.41±0.09	0.014±0.006
SVM	0.94±0.19	0.019±0.016
Tree	0.13±0.32	0.015±0.011

## Data Availability

Not applicable.

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
