# Peer review of "A Classification Method for Workers’ Physical Risk"

_sensors, 2023, doi:10.3390/s23031575_

Round 1
Reviewer 1 Report
This paper compares different classification approaches to identify risk conditions of workers to prevent accidents such as falls, slips, and fainting based on various physiological measurements. This research is important, and the paper is generally well written and clear. However, several issues must be addressed before considering publication. Here are more specific details.
Could you justify or explain the choice of a time window of 2.5 seconds? Why not 3 seconds, 5 seconds, or 1 seconds? You provide a reference ([32]) that might explain this choice but if this is the case you should summarize it in your paper.
What was the value of K in the ReliefK algorithm? And how did you choose it? Is it a sensitive parameter?
The description of the four machine learning algorithms is not sufficient. For instance, could you provide the values of all the parameters of these algorithms (e.g., value of K for K-NN, kernel type and regularization parameter for SVM, type of algorithm used to construct the decision tree etc.) and a reference to the software(s) used for them if you did not implement them by yourself.
Generally, the accuracy is not sufficient to correctly assess the performance of a machine learning algorithm (for instance see https://towardsdatascience.com/accuracy-is-not-enough-for-classification-task-47fca7d6a8ec), could you justify or explain the limitations of your choice and/or add other metrics such as precision, recall, F-score, sensitivity, specificity to complement the accuracy.
What is the total number of samples in your study? How many samples for each subject (for each feature)? This is not clear to me.
How do you deal with outliers (line 266, figure 4 etc.)? How are they detected? Are they used in the evaluation of the machine learning algorithms? Please clarify.
All these sensors seem cumbersome and annoying to wear for a worker in a realistic situation. Could you discuss this issue in your paper? If the worker does not want to wear them they will be useless isn’t it?
Some typos:
Line 55: coulde
Line 219: The participants
Line 231: The sentence “This first validation step serves to assess the actual need to use physiological information to identify risk perception” is repeated two times.
Line 240: “It is assessed” -> “It assessed”?
Line 268: whit -> with
Reviewer 2 Report
1. The introduction describes the application of SVM, why not mention the application of other three methods (kNN, LDA, Tree)?
2. Why use a 2.5s time window to extract statistical features? There is no clear explanation in the paper.
3. If these feature mentioned in the paper are added with other statistical feature, can the accuracy of the model be improved?
4. Can we consider the influence of different values of k on the prediction results?
5. The main parameters of each model should be mentioned in the paper so that the result/replication can be reproduced.
6. Can the time of training and prediction in Section 3 be tabulated or figure, so as to more intuitively compare the running speed of each model?
6. Can the training and prediction times in Section 3 be made into tables or figures to compare the running speed of each model more visually?
7. In Figure. 8, does the optimal set of physiological features proposed in this paper have universality? It is recommended that perform several experiments before arrive at an experiment.
